# Platinum Group Metals Nanoparticles in Breast Cancer Therapy

**DOI:** 10.3390/pharmaceutics16091162

**Published:** 2024-09-03

**Authors:** Sibusiso Alven, Sendibitiyosi Gandidzanwa, Basabele Ngalo, Olwethu Poswayo, Tatenda Madanhire, Blessing A. Aderibigbe, Zenixole Tshentu

**Affiliations:** 1Department of Chemistry, Nelson Mandela University, Gqeberha 6001, South Africa; s211175897@mandela.ac.za (S.G.); s217756352@mandela.ac.za (B.N.); s215054121@mandela.ac.za (O.P.); zenixole.tshentu@mandela.ac.za (Z.T.); 2Department of Chemistry, University of South Africa, Johannesburg 1710, South Africa; 3Department of Chemistry, University of Fort Hare, Alice 5700, South Africa; baderibigbe@ufh.ac.za

**Keywords:** platinum group metals, nanoparticles, breast cancer, drug delivery, cancer diagnosis and cancer therapy

## Abstract

Despite various methods currently used in cancer therapy, breast cancer remains the leading cause of morbidity and mortality worldwide. Current therapeutics face limitations such as multidrug resistance, drug toxicity and off-target effects, poor drug bioavailability and biocompatibility, and inefficient drug delivery. Nanotechnology has emerged as a promising approach to cancer diagnosis, imaging, and therapy. Several preclinical studies have demonstrated that compounds and nanoparticles formulated from platinum group metals (PGMs) effectively treat breast cancer. PGMs are chemically stable, easy to functionalise, versatile, and tunable. They can target hypoxic microenvironments, catalyse the production of reactive oxygen species, and offer the potential for combination therapy. PGM nanoparticles can be incorporated with anticancer drugs to improve efficacy and can be attached to targeting moieties to enhance tumour-targeting efficiency. This review focuses on the therapeutic outcomes of platinum group metal nanoparticles (PGMNs) against various breast cancer cells and briefly discusses clinical trials of these nanoparticles in breast cancer treatment. It further illustrates the potential applications of PGMNs in breast cancer and presents opportunities for future PGM-based nanomaterial applications in combatting breast cancer.

## 1. Introduction

Cancer remains a deadly threat to human health [1]. Globally, many lives are lost to various types of cancers [2]. In females, breast cancer is the most prevalent [3]. In 2020, over 2.3 million new cases were reported, and about 680,000 women died from breast cancer [3]. These numbers continue to increase each year, making it crucial to curb the burden of breast cancer [3]. By 2040, an estimated 1 million deaths and 3 million new cases are predicted worldwide [3].

The incidence of breast cancer increases with age. Various modifiable factors like physical inactivity, obesity, and alcohol consumption, as well as non-modifiable factors such as genetic mutations and family history, influence cancer risk [4]. Breast cancer often begins in the ducts (ductal carcinoma) or the lobules (lobular carcinoma) [1] and can be localised (in situ) or invasive (spreading to nearby tissues). Risk factors include hormonal influences, reproductive history, and lifestyle choices [4].

Breast cancer is caused by mutations in the progesterone and oestrogen receptors and develops in stages: Stage 0, I, IA, IB, II, IIA, IIB, III, IIIA, IIIB, and IV [5]. Each stage requires different treatment approaches and presents various challenges, such as low specificity, rapid clearance, dose-limiting toxicity, increased drug resistance, lower cancer cell pH, and difficulty eradicating cancer stem cells [3]. However, nanomedicine can circumvent these issues through targeted therapy, extending blood circulation time, regulating drug release, and inducing passive and active targeting [5].

Breast cancer exhibits significant tumour heterogeneity, leading to differences in clinical treatment outcomes and prognosis among patients [6]. Its high propensity to metastasise and develop drug resistance makes it challenging to cure [7]. Early diagnosis and treatment are crucial for improving survival rates. Nanomedicine can manipulate breast implants, target breast cancer cells, and induce immunomodulation and endothelial leakiness [8].

Developing countries face a higher impact, and current therapies have challenges like multidrug resistance, enhanced toxicity, nephrotoxicity, and low efficacy due to poor absorption, solubility, and delivery [9,10,11]. This highlights the need for more efficient therapies with minimal adverse effects [12]. Nanoparticles address significant problems in current cancer therapies, such as early cancer detection and toxicity. They can improve drug treatment efficacy by delivering higher concentrations to the target site, controlling the release mechanism, ensuring specificity and selectivity, and avoiding toxicity due to easy bodily clearance (Figure 1) [13].

Functionalising nanoparticles ensures effective, targeted, specific, and non-invasive cancer therapy. Cancer cells overexpress specific receptors, and nanomaterials can be functionalised to target these receptors [14]. Recent studies have shown the ability to preferentially deliver nanoparticles to tumour cells by targeting overexpressed receptors in breast cancer cells, such as human epidermal growth factor receptor 2 (HER2) and folate-functionalised nanoparticles [15].

Nanoparticles in cancer therapy continue to garner interest [16]. Metallic nanoparticles show excellent prospects in targeted therapy [6] and multiple biomedical applications [17]. PGMs, like noble metals, can be tailored to meet bionanomaterial requirements for diagnosis [18], therapy [19], drug delivery [20], biosensing, and imaging [21,22]. There is potential for overcoming current therapeutic challenges and improving diagnosis, detection, and therapy through PGMs [23]. Limited research has been conducted on PGMs in breast cancer therapy.

This review discusses the prospects of PGMs—platinum (Pt), palladium (Pd), rhodium (Rh), ruthenium (Ru), iridium (Ir), and osmium (Os)—nanoparticles in breast cancer therapy and provides an overview of their use in cancer therapy, specifically breast cancer.

## 2. Nanotechnology in Breast Cancer Therapy

Nanomedicine can be used in breast cancer therapy. Preferred nanomaterials should be non-toxic, biocompatible, and small in size. They should have effective capping or stabilising agents, remain stable in physiological environments and maintain good circulation time [24]. Additionally, they must exhibit strong targeting, selective targeting, and efficient clearance [25]. Despite challenges, nanomedicine offers better alternatives to current cancer procedures [26].

Nanomedicines can use passive and active targeting of cancer cells and stimuli-responsive tumour targeting [27]. Extensive research is being conducted on liposomes, polymeric micelles and nanoparticles, dendrimers, and carbon nanotubes [15]. Passive and active targeting using nanomaterials has become a significant research area [28]. There are concerns about the use of heavy metals in the body [24]. Therefore, ideal PGMs should exhibit better therapeutic outcomes, improved bioavailability [29], prolonged survival, reduced side effects [30], and preferably greener synthesis [26]. Despite challenges, recent research shows nanomaterials as promising solutions to these obstacles [20].

Nanomaterials can be used as drug-delivery systems in therapy, diagnosis, and imaging [31]. Many nano-formulations are in clinical trials for cancer treatment [1], but physiological and pathological variations between animal models and humans create a translational gap. Additionally, challenges like nanotoxicity, nanomedicine complexity, incomplete understanding of nano-tumour interactions, reproducible and scalable nanoparticle synthesis, cost, and adherence to regulations need to be overcome for improved nanomaterials in cancer therapy [16,32,33].

Over ten clinical drugs are available for breast cancer [25]. Some breast cancer nanomedicines in clinical trials include NK-105, SPI-077, and Xyotax^®^ for metastatic breast cancer, and Genexol^®^-PM, Mitoxantrone hydrochloride liposome, and Rexin-G for recurrent or metastatic breast cancer, all administered through intravenous infusion. Doxil^®^ was approved by the Food and Drug Administration (FDA) in 1995, Abraxane^®^ in 2005 and 2008 by the European Medicines Agency (EMA), and Myocet^®^ in 2000 [4]. Clinically approved anticancer drugs for breast cancer treatment include Myocet, Apealea, Marqibo, Vyxeos, and Oncaspar [33]. Very few nanomedicines are available for the clinical treatment of breast cancer. These include Nanoxel, Doxil^®^/Caelyx^®^, Lipusu^®^, Kadcyla^®^, Lipodox^®^, and Abraxane^®^, highlighting an urgent need to develop more nanomedicines that can reach clinical trials [15]. Several breast cancer nanomedicines are currently under clinical trials (Table 1). These nanomedicines are in various forms, including nanoparticles, dendrimers, micelles, non- and PEGylated liposomes, etc. (Figure 2).

Genexol^®^-PM, a formulation designed in the form of micelles to enhance the water solubility of paclitaxel, demonstrates a 25% higher water solubilisation efficiency than Taxol^®^. This nanosystem also shows a better maximum tolerated dose than Taxol^®^, decreased myelosuppression, excellent inhibition of P-glycoprotein, and higher cellular internalisation than Taxol^®^. Currently, Genexol^®^-PM is in clinical applications in Bulgaria, Hungary, and South Korea and is in Phase II clinical trials in the USA [34]. The promising results of phase II clinical studies conducted by Lee et al. [35] show significant efficacy of Genexol-PM^®^ at 300 mg/m^2^ administered every 21 days in first-line or anthracycline-pre-treated patients with metastatic breast cancer (MBC). The total response rate of Genexol-PM^®^ in patients was approximately 58.5%, which is higher than the response rate of about 47.6% reported by ABI-007 (Abraxane; American BioScience Inc, New York, NY, USA.) at the same dosing regimen. Moreover, Genexol-PM^®^ also exhibited superior efficacy against MBC compared to conventional paclitaxel [35], indicating a bright future for this nanosystem in breast cancer therapy.

Phase II clinical trials investigating the efficacy and safety of Xyotax^®^, also known as CT-2103 (poly-L-glutamate-paclitaxel conjugate), in patients with HER2-negative MBC proved that Xyotax^®^ was effective in women with MBC [36]. However, the study was terminated due to unexpected neurotoxicity and hypersensitivity reactions [36]. Baselga et al. conducted phase III clinical studies of Myocet^®^, paclitaxel and trastuzumab in human epidermal growth factor receptor 2 (HER2)-positive MBC patients [37]. These clinical trials demonstrated that Myocet^®^ in combination with paclitaxel and trastuzumab and combination therapy of paclitaxel and trastuzumab resulted in median progression-free survival (PFS) of 16.1 and 14.5 months, respectively, with a hazard ratio (HR) of 0.84. In patients with progesterone receptor (PR)- and oestrogen receptor (ER)-negative breast tumours, PFS was 14.0 and 20.7 months, respectively, with HR 0.68. These findings revealed no significant clinical improvement in the combination therapy regimen of paclitaxel and trastuzumab combined with Myocet^®^ [37]. This underscores the need for further research and development in the field of breast cancer therapy.

Rugo et al. evaluated the efficacy and safety of Lyso-thermosensitive liposomal doxorubicin (LTLD, Thermodox^®^) in partial phase I/II clinical studies in 17 patients with breast cancer recurrence at the chest wall [38]. These studies showed that Thermodox^®^ resulted in a localised reaction in the treatment area in two patients consisting of pain, woody induration, and erythema that led to the termination of treatment. Furthermore, the clinical findings showed that Thermodox^®^ at a dose of 40 mg/m^2^, together with hyperthermia, provides an optimistic and patient-friendly treatment choice for patients with breast cancer recurrence at the chest wall [38]. Rosenbaum and co-workers conducted a phase I clinical trial of MAG-Tn3 for immunisation patients with localised breast cancer. The outcomes revealed that all vaccinated patients developed high levels of Tn-specific antibodies, which precisely detected Tn-expressing human cancer cells and destroyed them via a complement-dependent cytotoxicity mechanism [39]. These nanomedicines’ successes show an excellent opportunity for more specialised therapeutics containing platinum-group-based nanomaterials.

## 3. Platinum Group Metal Nanoparticles in Breast Cancer Therapy

Nanoparticles can kill cancer cells through various pathways, such as cell cycle arrest, DNA damage, apoptosis, and reactive oxygen species (ROS) generation [40]. Nanomaterials hold great promise for more effective and specific breast cancer treatment [41,42]. Despite the limited nanomedicine-based products on the market, using nanomaterials in cancer therapy warrants further exploration. PGM-based nanomaterials (and compounds) in cancer applications can be summarised below (Figure 3). The success of metal-based drugs like cisplatin in chemotherapy has generated significant interest in investigating the therapeutic potential of different metals for developing new therapeutic agents [43]. Among these, platinum group metals (PGMs) show considerable promise for breast cancer therapy due to their diverse biomedical applications (Figure 3).

### 3.1. Platinum-Based Nanoparticles

Platinum (Pt) plays an essential role not only in nanomedicine but also in various other areas. Platinum nanoparticles (Pt NPs) have significant biomedical applications, including bone loss and bone allograft, dentistry, drug delivery, antibacterial and antifungal properties, bioimaging, anti-inflammatory properties, nano-diagnostics, and photothermal and radiotherapy [44,45]. Cisplatin, carboplatin, nedaplatin, lipoxal, lobaplatin, and heptaplatin are some Pt compounds used in cancer treatment [44,46,47]. However, their use is limited by adverse effects such as ototoxicity, neurotoxicity, and nephrotoxicity [48].

Pt NPs can mitigate these side effects and play a crucial role in cancer treatment with improved anticancer activity against various cancers, including breast cancer. Pt NP synthesis methods include chemical methods (e.g., reduction, fusion approach, microemulsion, pyrolysis, sol-gel process), physical methods (e.g., solvothermal processes, laser ablation, inert gas condensation), and biological methods (green synthesis using microorganisms or plant extracts) [49]. Pt NP uptake via endocytosis or passive diffusion depends on size, incubation time, and concentration. Internalised Pt NPs induce cytotoxicity by disrupting chromosomal DNA strands, hindering replication, and stimulating cell cycle arrest and apoptosis. Pt NPs also release active free Pt^2^⁺ ions, inhibit hydroxyl radical generation, and affect cellular metabolic activity [50]. Pt NPs trigger apoptosis in cancer cells through multiple target-specific pathways [50].

Al-Radadi synthesised Pt NPs via green synthesis using *Saudi’s Dates* extract, showing high anticancer efficacy against breast cancer cells (MCF-7) with less than 40% cell viability at 400 µg/mL [51]. Sahin et al. demonstrated high inhibition of MCF-7 cell proliferation using pomegranate extract green-synthesised Pt NPs [52]. Jha et al. used ethanol extracts of *Punica granatum* fruit for green synthesis of Pt NPs, showing significant cytotoxic potential against MCF-7 and MDA-MB-231 cells compared to the positive control, Mitomycin-C [53]. Abed et al. formulated Pt NPs using *Ziziphus spina christi*, exhibiting high inhibition against MCF-7 cells and converting near-infrared radiation to heat to stimulate apoptosis [54]. Baskaran et al. employed *Streptomyces* sp. for green synthesis of Pt NPs, which showed good antitumour properties in MCF-7 cells [55].

Rokade et al. synthesised Pt and palladium (Pd) nanoparticles using Gloriosa superba tuber extract, demonstrating significant anticancer activity against MCF-7 cells [56]. Manzoor et al. used *Psidium guajava* plant extract for Pt NP synthesis, showing promising anticancer efficacy in breast cancer [57]. Puja et al. prepared spherical polyvinyl pyrrolidone (PVP)-modified Pt nanoparticles that significantly inhibited the growth of human MCF-7 breast cancer cell lines in a dose-dependent manner, with an IC_50_ of 96.36 μg/mL [58]. Ruiz et al. fabricated poly (lactic-co-glycolic acid) (PLGA)-functionalised Pt nanoparticles, exhibiting cell viability of 2% using MBA-MB-231 at 200 µg/mL after a 5-day incubation, indicating excellent anticancer activity against breast cancer cells [59]. Fu et al. created polyethylene glycol (PEG)-modified mesoporous Pt doxorubicin nanoparticles. Drug release studies at pH 5.5 (tumour microenvironment) and pH 7.4 demonstrated a controlled release mechanism of doxorubicin from Pt nanoparticles, with a faster release for 36 h at lower pH, revealing their potential for cancer therapy. Confocal imaging and TEM results showed prolonged doxorubicin release and high cellular internalisation of platinum nanoparticles in MCF-7/ADR cells, suggesting good anticancer activity against breast carcinoma [60]. Li et al. developed PEG-coated Pt nanoparticles conjugated with rutin that exhibited excellent cytotoxic effects against MCF-7 cells compared to non-tumourigenic breast cells (MCF-10A) [61].

Rashidzadeh et al. fabricated alginate-coated Pt NPs for cancer therapy and radiosensitisation of breast cancer tumours. In vivo studies of Pt NPs combined with X-ray radiation in female BALB/c mice bearing 4T1-breast tumours (~300 mm^3^) showed superior growth inhibition of breast tumours compared to pristine alginate-coated Pt nanoparticles or free X-ray radiation, suggesting a synergistic anticancer effect that could significantly contribute to breast cancer therapy [62]. Ramanathan et al. reported chitosan-stabilised platinum nanoparticles that exhibited good anticancer efficacy against breast cancer cells (MDA-MB-231) with an IC_50_ of 12 ± 2.14 μg/mL and excellent cytocompatibility towards human embryonic kidney cells (HEK-293) [63].

Mohammadi et al. synthesised Pt NPs efficiently taken up by targeted cancer cells [64]. In vitro cytotoxicity studies using MTT assay showed that Pt NPs possessed excellent cytotoxic effects against MCF-7 cells with an IC_50_ of 6.8294 mg/mL, revealing their efficacy in breast cancer treatment [64]. In vivo studies using 4T1-breast cancer cell-implanted nude mice reported by Wang et al. demonstrated induced lung metastasis at high doses (10 mg/kg) in a dose-dependent manner [65]. Lee et al. reported platinum@gold nanoparticles with excellent anticancer properties against MCF-7 cells [66]. Patel et al. synthesised Pt nanoparticles conjugated with doxorubicin for breast cancer treatment. The MTT assay showed IC_50_ values of 2.3 and 0.3 μg/mL against MCF-7 cells for PVP-Pt nanoparticles and doxorubicin-Pt nanoparticles, respectively. The IC_50_ values were 4.6 and 1.0 μg/mL against MDA-MB-231 cells for PVP-Pt nanoparticles and doxorubicin-Pt nanoparticles, respectively, demonstrating better cytotoxicity of doxorubicin-Pt NPs against breast cancer cells [67].

In summary, Pt-based NPs are particularly notable for their multifaceted properties that make them suitable for various biomedical uses, including diagnostics, drug delivery, therapy, and imaging [49]. These nanoparticles also exhibit antibacterial [68], antimicrobial [62], antiviral [69], and anti-inflammatory properties, making them potential candidates for combination therapies [68]. They are also excellent photosensitisers [70], biosensors [71], imaging agents in computed tomography [69], and radiation dose enhancement agents [69]. These studies highlight the growing interest in using platinum nanoparticles in cancer therapy.

### 3.2. Palladium-Based Nanoparticles

Palladium nanoparticles (Pd NPs) are a valuable tool for cancer therapy due to their excellent catalytic activity and high surface area, making them suitable for various therapeutic applications [72]. Depending on the synthesis method and intended applications, tailored sizes ranging from 1 to 100 nm are possible, enabling multiple roles in cancer therapy [71]. The biomedical applications of palladium nanoparticles (Figure 4) range from gene therapy to photodynamic therapy and targeting strategies [71].

Gandidzanwa et al. [14] showed that Pd NPs could be functionalised to target only cancer cells. A proof-of-principle study for developing palladium-based nano-radiopharmaceuticals using palladium-based nanoparticles used non-cytotoxic concentrations of palladium-based nanoparticles. The study followed the distribution and accumulation of folate-functionalised Pd NPs in breast cancer cells, using four breast cancer cell lines (MDA-MB-231, MDA-MB-468, MCF-7) and the non-tumourigenic MCF-10A cell line as a control. At 10 µM palladium content, the nanoparticle systems were non-cytotoxic. However, at higher concentrations (100 µM palladium content), the Pd-based nanoparticles exhibited anticancer properties towards the cell lines, showing successful functionalisation, characterisation, and accumulation of folate-functionalised Pd-based nanoparticles [14].

Alkhalidi et al. [73] reported the green synthesis of acid protease-mediated palladium nanoparticles (ACPs-PdNPs) for biomedical applications in MCF-7 cells. The synthesised NPs inhibited MCF-7 cells with an IC_50_ of 66.37 µg/mL. Enhanced penetration and bioavailability of the small-size NP allowed them to penetrate bacterial biofilms, improving the efficacy of antibacterial treatments [73]. These approaches have demonstrated significant and practical outcomes, indicating their potential for completely eradicating cancer cells. PdNPs exhibit robust localised surface plasmon resonance (LSPR) when exposed to near-infrared (NIR) light [74], enabling selective cancer cell destruction. They act as photosensitisers upon light activation, generating reactive oxygen species (ROS) and causing stress to cancer cells, inhibiting their growth and multiplication around infected areas [75,76]. Recent developments and synthesis of Pd-based hybrid nanoparticles modified with chitosan oligosaccharide (COS) and functionalised with arginylglycylaspartic acid (RGD) peptide (Pd@COS-RGD) have shown good biocompatibility and physiological stability in breast cancer cells [77].

In a study by Yuan et al. [78] focusing on MDA-MB-231 cells, tubastatin A (TUB-A) emerged as a promising candidate for achieving potent anticancer effects when combined with Pd NPs. The research highlighted that the synergistic effect of TUB-A functionalised with Pd NPs (2.7 nmol/mg of protein) resulted in more significant cytotoxicity against cancer cells compared to individual treatments of TUB-A (1 nmol/mg of protein) or Pd NPs (1.7 nmol/mg of protein) alone [78]. Combining these nanostructures significantly reduced cell viability by up to 70%, whereas TUB-A alone achieved a 25% reduction, and Pd NPs alone achieved a 25% reduction. This enhanced cytotoxicity was supported by observable changes in cell morphology, contributing to reduced toxicity overall [78].

Shahriari et al. [79] developed site-specific immobilisation of Pd NPs over biodegradable chitosan/agarose modified ferrite NP (Fe_3_O_4_@CS-Agarose/Pd) against human breast cancer. The cell viability of Fe_3_O_4_@CS-Agarose/Pd nanocomposites was measured at concentrations (0–1000 μg/mL) for all breast adenocarcinoma (MCF-7), breast carcinoma (Hs 578Bst), and infiltrating ductal cell carcinoma (Hs 319.T). The findings on IC_50_ of Fe_3_O_4_@CS-Agarose/Pd nanocomposite and BHT in the antioxidant test at concentrations (μg/mL) were 131 and 159, respectively, and MCF-7, Hs 578Bst, Hs 319.T, and MDA-MB-453 were reported to be 347, 250, 188, and 519 at the same concentrations [79]. The highest cell viability percentage of nanocomposite against normal (HUVEC) cell line was at 0 (μg/mL) concentration, while the lowest cell viability percentage was at 1000 (μg/mL) concentration. The same trend of concentration (0–1000 μg/mL) decreasing from lowest to highest concentration value was observed for the anti-human breast cancer properties from lowest to highest concentration against MCF-7, Hs 578Bst, Hs 319.T, and MDA-MB-453 cell lines [79].

Pd NPs functionalised with precise ligands transported therapeutic agents that disrupt irregular DNA replication and protein synthesis, potentially impeding abnormal cancer cell proliferation [80]. Wu et al. [81], in their report for anticancer treatment, successfully synthesised effective manganese-oxide and palladium nanoparticle-co-decorated polypyrrole/graphene oxide (MnO_2_@Pd@PPy/GO) nanocomposites targeting cancer cells via photothermal therapy (PTT) and chemodynamic therapy (CDT). The highest % viability of cancer cells for PTT was more than 90%, followed by CDT, which was 74.7% at pH 5.0. All these findings show great potential for inhibiting cancer cell irregular growth.

Vaghela et al. [82] green-synthesised Pd NPs and showed excellent stability and efficacy against MCF-7 cells. Dose-response testing of the synthesised PdNPs showed a 60% inhibition of cancer cell activity. Ghosh et al. [82] reported using *Dioscorea bulbifera* tuber extract to produce platinum–palladium bimetallic NPs ranging in size from 10 to 25 nm. The nanoparticles were centrifuged at 10,000 rpm at room temperature and incubated at 25 °C for 2.5 h at pH 7.4, resulting in over 70% cell death. Fahmy et al. [83] also noted the anticancer properties of Pd and Pt NPs against MCF-7 cells. Rokade et al. [84] synthesised Pd NPs that showed significant antiproliferative activity on MCF-7 cells, achieving an anticancer activity of 36.26  ±  0.91% in a similar study [56].

Abdel-Fattah et al. [85] used extracts from *Prunus amygdalus* and blackberry fruits to synthesise Pd NPs targeting MCF-7 cells. Fahmy et al. [86] explored the anticancer efficacy of Pd NPs synthesised from *Santalum album*, *Cinnamomum camphora*, soybean leaves, and *Solanum nigrum* (Figure 5).

A comprehensive assessment of Pd NPs for breast cancer treatment requires diverse studies, including cytotoxicity, genotoxicity, in vivo experiments, surface modifications, and safety evaluations. Safarkhani et al. [87] explored drug delivery efficacy using nanocomposites like rGO@MWCNT and @DOX/PEG, constructed with (AgL_2_ or PdL_2_) complexes. The research evaluated biocompatibility and suitability in breast cell lines (MCF-10A, MCF-7, and MDA-MB-231), showing significant cytotoxicity against cancer cells with high biocompatibility with normal cell lines [87]. Evaluations were conducted at pH levels 7.4 and 5.5, with treatments analysed after 24 h. The in vitro drug release was less than 20%, attributed to PEG incorporation, highlighting therapeutic potential with minimal risk to normal cells [87].

Bharathiraja et al. [77] investigated the biocompatibility and effectiveness of Pd@COS-RGD and Pd@COS nanoparticles for MDA-MB-231 cells using in vitro and in vivo models. MTT assays assessed cell viability at concentrations ranging from 0 to 50 ppm, showing significant cell viability (>75%) with Pd@COS NPs > Pd@COS-RGD > Pd-CTAC. Pd@COS-RGD demonstrated the highest mass of Pd-NPs per cell (>70 pg) at 4–6 h intervals [77]. The analysis showed increased cancer cell death rates without adversely affecting normal cells, underscoring the potential of palladium nanoparticles combined with laser technology to inhibit breast cancer cell growth [77].

Pd nanomaterials exhibit excellent photothermal stability and are highly stable. Functionalising palladium nanoparticles with polymers improves their biocompatibility and stability [84]. Functionalizing Pd NPs with chitosan and RGD peptide for Pd@COSRGD enabled their use in laser imaging and tumour diagnosis [88]. PdAu NPs, like PtAu NPs, are potential radiosensitisers in photothermal therapy for colorectal cancer, significantly inhibiting cancer cell proliferation and viability [89]. Pd NPs functionalised with trimethyl-chitosan (TMC) showed improved biocompatibility, accumulation, and stability in MDA-MB-231 cells. TMC/PdNPs produced a photothermal effect when irradiated with an 808 nm NIR laser, showing significant cytotoxic effects in 2D and 3D MDA-MB-231 models. TMC/PdNPs are excellent PTT agents [72].

PdNPs showed promise as biosensors. Pd NP-dopamine biosensors exhibited high sensitivity, selectivity, and a wide detection range [90]. At higher concentrations, PdNPs inhibit cell proliferation in MCF-7 cell lines [90]. PVP-Pd NPs displayed anticancer activity against MCF-7 cells [91]. Homogeneous Pd NPs induced apoptosis in colorectal cancer cells more effectively than cisplatin [92]. Pd nanoparticles have various biomedical applications [90]. Palladium nanoplates are useful for therapeutic gene loading and release in vitro combinational cancer therapy [93]. Pd NPs also served as drug delivery vehicles, delivering ^131^I, an anticancer radioisotope used in radiotherapy, and doxorubicin (a chemotherapy drug), producing synergistic anticancer effects through combined chemo-, photothermal-, and radiotherapy in breast cancer cell lines (MCF-7), as demonstrated by Song et al. [94]. Gil et al. reported the tri-modal use of palladium nanoparticles synthesised from chaga in HeLa cells [95].

Pd NPs can be used in PTT by functionalising them with RGD peptides, which showed better cytotoxicity in MDA-MB-231 cells [77]. This indicates promising future treatments for Pd NPs in theranostic breast cancer. Palladium nanoparticles exhibit significant biomedical applications comparable to silver and gold nanoparticles [95]. Phan et al. discussed the intrinsic properties of palladium nanoparticles for biomedical applications like imaging and therapy [59]. Pd nanoparticles also showed antioxidant and anti-tyrosinase activities [96], with shape not affecting their anticancer properties [97]. Bugwandeen et al. reported Pd NP’s anticancer properties for breast cancer (MCF-7) [98], with Pd NPs functionalised with doxorubicin exhibiting near-infrared triggered PTT and PDT synergism in killing cancer cells [99]. In vitro and in vivo PDT studies of Pd NPs showed excellent medical prospects [100].

In summary, Pd-based NPs have also been extensively studied for their potential as biosensors [101], prodrugs, and prodrug activators [102], as well as in radiotherapy [94], PTT and PDT [75,97], SPECT, CT, and MRI imaging [103], and gene/drug carriers [93]. Like Pt NPs, their antimicrobial [104], anticancer [105], and drug delivery capabilities [106] make them appealing candidates for combinatorial therapy research. However, the full potential of palladium nanoparticles is yet to be explored.

### 3.3. Ruthenium-Based Nanoparticles

Starvation therapy, a novel approach in cancer treatment, disrupts tumour energy supply, effectively inhibiting tumour growth [107,108,109,110]. Rapidly proliferating cancer cells have a high glucose demand, making them vulnerable to glucose deprivation. Glucose oxidase (GOx) catalyses glucose oxidation, initiating starvation therapy by depleting glucose and inducing cancer cell apoptosis. Tumour hypoxia, however, limits glucose oxidase efficacy.

Nanonzymes, nanomaterials with catalytic activity, offer a promising solution [107]. Iridium and ruthenium nanoparticles (IrRu NPs) function as nanozyme reactors, exhibiting catalase and peroxidase-like activities. By catalysing glucose oxidation and converting H_2_O_2_ into toxic species under acidic conditions, IrRu NPs enable dual-stage enzymatic reactions within the tumour microenvironment [107]. IrRu-GOx@PEG NPs, synthesised with IrRu NPs and GOx encapsulated in polyethylene glycol (PEG), exhibit dual enzyme activity. In vitro studies showed efficient glucose degradation and ROS generation, inducing apoptosis in 4T1 cancer cells (mouse breast cancer cells). These nanoparticles penetrate 3D tumour cell clusters, demonstrating superior tumour permeability compared to conventional nanoparticles [107].

In vivo, these nanoparticles accumulate in tumour sites, showing minimal toxicity and maintaining glucose homeostasis. In vivo studies on 4T1 tumour-bearing mouse models demonstrated the efficacy of IrRu-GOx@PEG NPs in inhibiting tumour growth through synergistic starvation and oxidative therapies [107]. Histopathological analyses revealed extensive tumour necrosis and reduced tumour proliferation, indicating the therapeutic potential of IrRu-GOx@PEG NPs in combating breast cancer metastasis [107]. These findings highlight the potential of nanozyme reactors as a promising strategy in breast cancer treatment, warranting further clinical exploration.

Ruthenium(II) polypyridyl functionalised selenium nanoparticles (Ru-SeNPs) (Figure 1) have shown remarkable efficacy in inhibiting cancer cell proliferation in MCF-7 cells [111]. The IC_50_ value of Ru-SeNPs against MCF-7 cells was 20.2 ± 2.3 µg/mL, notably lower than cisplatin (24.4 ± 1.9 µg/mL), highlighting their potent cytotoxic effects. Compared to selenium nanoparticles (SeNPs) and the ruthenium polypyridyl complex (RuBP) with IC_50_ values of 42.9 ± 3.5 and 43.5 ± 2.6 µg/mL, respectively, Ru-SeNPs exhibited significantly enhanced cytotoxicity against MCF-7 cells [111]. This substantial increase in cytotoxicity underscores the efficacy of ruthenium nanoparticles as a promising alternative or complement to conventional chemotherapy agents like cisplatin. More studies are necessary to realise the therapeutic benefits of ruthenium nanoparticles in breast cancer management.

Exploring green synthesis methods for producing nanoparticles, particularly ruthenium oxide nanoparticles (RuONPs), offers an environmentally friendly and cost-effective approach with potential applications in breast cancer therapy [112]. Mfengwana & Sone synthesised RuONPs using green methods and evaluated their cytotoxicity against MCF-7 breast cancer cells and Vero cells (non-cancerous) using the MTT assay [112]. Despite successful synthesis and characterisation, RuONPs showed low anticancer efficacy against MCF-7 cells, suggesting limited potential as a standalone anticancer drug. This low efficacy suggests further optimisation, chemical modification, and exploration of alternative applications, such as using RuONPs as drug carriers or in combination therapies. Ruthenium has also been used in photodynamic therapy [113]. Transferrin-modified ruthenium NPs showed excellent biocompatibility and photothermal properties in cancer treatment [114]. Ruthenium(II) tris(bipyridyl) cationic complex (Ru(bpy)_3_^2+^) incorporated into UiO-67 nanoscale metal-organic frameworks exhibited good biocompatibility, cytotoxic rates, and dual imaging and photodynamic therapy abilities [115].

Ruthenium compounds possess unique biochemical characteristics and inhibit tumour growth [116]. However, some Ru complexes exhibited poor targeting, non-selectiveness, unintended toxicity, and bioavailability issues [117]. Despite these challenges, ruthenium agents have been reported to be easily cleared from the body [118]. Ru NPs possess excellent chemodynamic, photothermal, and photodynamic activities [119]. More studies are required to expand on these promising findings [120].

Ru-based compounds have shown potential applications in various biomedical fields [121], including their use as nanodrugs [116], anticancer agents [122], antibacterial [123], antimicrobial, antiviral, and antiparasitic agents [124], as well as anti-inflammatory [43] and antimetastatic chemotherapeutic agents [125]. They are also being investigated as potential anti-SARS-CoV-2 agents [124]. With BOLD-100 in clinical trials [126], Ru-based compounds are gaining attention as photosensitisers in PTT [114] and PDT [15,36], chemotherapy agents [127], imaging agents [128], biosensors [129], and in drug delivery systems [15,29,33]. There is a need for further research on ruthenium-based nanomaterials.

### 3.4. Rhodium-Based Nanoparticles

Rhodium (II) citrate and maghemite NPs exhibited antitumour effects against 4T1 MBC cells [130]. Rhodium citrate inhibited DNA synthesis and proliferation in MDA-MB-231 and MCF-7 cells [131]. Rhodium NPs (RhNPs) induced apoptosis via PDT as a photosensitising agent [132]. Rh NPs have low toxicity, and their use has been proven at transcriptome and metabolome levels [133]. Rhodium (III) based compounds showed good anticancer properties and prospects in cellular imaging [134] and the treatment of triple-negative breast cancers [135]. Rh(III) arenyl showed anticancer properties in vivo [136], and rhodium-based nanozymes have shown promise [137].

Research by da Silva Nunes et al. explored the potential of citrate-coated rhodium(II) iron oxide nanoparticles [138]. This research introduced novel superparamagnetic iron oxide (SPIO) nanoparticles stabilised by rhodium(II) citrate, enhancing their therapeutic action [138,139,140]. The study investigated colloidal stability, intracellular accumulation in breast tumour cells, and the effect of different dispersion media on nanoparticle stability and behaviour [138].

Rhodium(II) citrate, Rh_2_(H_2_cit)_4_, adsorbed onto maghemite (Magh) nanoparticles in a pH-dependent manner, with maximum adsorption at pH3 [138]. The adsorption isotherm indicated a strong interaction between the complex and the nanoparticle surface. Stability was similar in saline (phosphate-buffered saline) but decreased in foetal bovine serum (FBS) after 20 days. TEM studies revealed that Rh_2_(H_2_cit)_4_-coated nanoparticles were taken up by 4T1 breast tumour cells, inducing distinct cell responses compared to citrate-functionalised nanoparticles. Magh-Rh_2_(H_2_cit)_4_-250 nanoparticles induced tissue necrosis and were present in the cytoplasm and nucleus of tumour cells [138]. This suggests the potential cytotoxic effect of Rh(II) citrate-coated SPIO nanoparticles against breast cancer cells, making them promising for drug delivery and therapy.

To address systemic toxicity from metal complexes in cancer treatment, Chaves et al. proposed combining dirhodium citrate with magnetic nanoparticles [141]. A study comparing free Rh_2_(H_2_cit)_4_, Rh_2_(H_2_cit)_4_-loaded maghemite NPs (Magh-Rh_2_(H_2_cit)_4_), and maghemite nanoparticles loaded with citrate (Magh-cit) found that Magh-Rh_2_(H_2_cit)_4_ and Magh-cit induced apoptosis mediated by ROS independently of caspase 3 expression. Rhodium citrate with maghemite nanoparticles induced apoptosis in MCF-7 cells. These nanoparticles triggered cytochrome C release, caspase activation, and DNA fragmentation. Cytotoxicity assays showed enhanced potency of Magh-Rh_2_(H_2_cit)_4_ compared to free Rh_2_(H_2_cit)_4_, with minimal impact on non-tumour cells. ROS production was significantly increased by Magh-Rh_2_(H_2_cit)_4_, implicating oxidative stress in cell death [141].

In another study, Rh(II) citrate-loaded maghemite nanoparticles (Magh-Rh_2_Cit) showed promising potential in breast cancer therapy, demonstrating cytotoxic effects on 4T1 and MCF-7 cell lines and inhibiting tumour growth in animal models [130]. Evaluating the systemic antitumour activity and toxicity of free Rh_2_Cit and Magh-Rh_2_Cit in mice bearing orthotopic 4T1 breast carcinoma, both formulations significantly reduced tumour volume compared to controls (Figure 6a) [130,142]. Magh-Rh_2_Cit decreased tumour area without inducing hematotoxicity or elevating serum ALT and creatinine levels, indicating low liver and kidney toxicity (Figure 6b) [130]. Histopathological examination supported the lack of toxicity induced by rhodium-based nanoformulations, confirming their safety profile.

The enhanced antitumour effect of Magh-Rh_2_Cit may be due to Rh(II) citrate’s hydrophilicity, improving its body dispersion and accumulation in tumour tissues [130]. Combining nanoparticles with Rh_2_Cit may facilitate passive tumour targeting via the EPR effect, increasing drug bioavailability and tumour-specific delivery [130]. Citric acid’s antioxidant properties and glycolysis inhibition may contribute to the antineoplastic activity of the nanoparticles [130,143,144]. Histological analysis confirmed the malignant nature of 4T1 tumours and demonstrated the efficacy of Rh_2_Cit and Magh-Rh_2_Cit in inducing tumour necrosis (Figure 6c,d) [130]. This study represents a crucial step in understanding rhodium-based nanoparticles’ role in combating breast cancer.

Studies have shown that Rh citrate-associated maghemite nanoparticles induce cytotoxicity in human tumour cells within 24 h of exposure, with varying effects over time [131]. While nanoparticles displayed a time-dependent cytotoxic effect on metastatic MDA-MB-231 cells, non-metastatic MCF-7 cells’ viability seemed to recover after 48 h, possibly due to resistance mechanisms mediated by P-glycoprotein (PGP) proteins. The heightened responsiveness of MDA-MB-231 cells to Rh citrate-associated maghemite nanoparticles compared to MCF-7 cells highlighted the nanoparticles’ potential in targeting metastatic cells [131]. Additionally, nanoparticle treatment reduced metastatic cells’ migratory capacity and induced S phase arrest in the cell cycle, indicating interference with DNA synthesis and proliferation. However, potential resistance mechanisms and differential cell behaviours over time indicate the need for further investigation and optimisation of treatment strategies. The findings emphasise understanding the underlying mechanisms of action, including PGP protein interactions and cell cycle arrest induction [131].

Researchers have discovered that Rh_2_(H_2_cit)_4_ with magnetic nanoparticles, such as maghemite nanoparticles and magnetoliposomes (Lip-Magh), enhances cytotoxic potency and specificity towards MCF-7 and 4T1 cells [145]. Rh_2_(H_2_cit)_4_-loaded Magh and Lip-Magh nanoparticles induced significantly higher cytotoxicity in cancer cells than free Rh_2_(H_2_cit)_4_, with 4.6 times potency observed. This enhanced cytotoxicity is more selective towards carcinoma cells, indicating potential for targeted delivery and reduced toxicity to normal breast cells (MCF-10A). Observed morphological alterations and increased annexin-V expression suggested the involvement of apoptosis and autophagy [145].

Rh NPs have potential in cancer phototherapy [146] and exhibit excellent antitumour properties [142]. They have also been used as nanozymes in biomedical therapy and biosensing applications [137]. Additionally, rhodium nanoparticles are employed as bioimaging agents [147], photosensitisers in PDT, photothermal agents, and radiosensitisers. Rhodium nanoparticles can be further explored in breast cancer therapy studies.

### 3.5. Osmium-Based Nanoparticles

Research on osmium nanoparticles for breast cancer treatment is sparse [148], primarily due to their scarcity, toxicity concerns, and challenges in characterisation owing to their small size (<2 nm) [149]. Current studies predominantly focus on osmium complexes rather than nanoparticles, showcasing their potential in addressing gaps in current breast cancer therapies, particularly against aggressive secondary and tertiary tumours [150]. For instance, Suntharalingham et al. investigated the selective toxicity of osmium (IV) nitrido complexes against breast cancer stem cell-enriched populations, specifically targeting CD44-positive cancer stem cells [149]. This complex mimics the mechanism of action of salinomycin, which effectively targets cancer stem cells and may overcome challenges such as chemotherapy resistance, metastasis, and tumour recurrence [149].

Compared to traditional platinum-based anticancer agents, osmium nitrido complexes show promise in actively inhibiting cancer cells [151,152]. Huang et al. demonstrated the anticancer activity of osmium (IV) nitrido complexes in HepG2 cells, inducing apoptosis via mitochondrial pathways and G2/M phase cell cycle arrest [153]. These complexes modulate protein homeostasis, upregulating proteins associated with DNA metabolism while downregulating oxidative stress-related proteins, highlighting their potential therapeutic value [153]. Additionally, studies by Needham et al. illustrated the cytotoxicity of ^131^I-labelled arene osmium (II) complexes in MCF-7 cells, underscoring their efficacy in vivo and potential as therapeutic agents [154].

However, several challenges must be addressed before osmium nanoparticles can be considered for breast cancer treatment [155]. These include toxicity issues, particularly the oxidation of osmium to the highly toxic OsO_4_ compound, and techno-economic concerns to ensure affordability and accessibility [156]. Further research is imperative to investigate these aspects comprehensively and explore alternative sources of osmium to facilitate the implementation of osmium nanoparticles in breast cancer therapy.

Despite limited research, osmium-based compounds demonstrate significant potential for future applications in biosensing [157], nanozymes [158], chemotherapy [158], immunotherapy [158], photothermal [158] and photodynamic therapy [158], and combinatorial cancer therapy [158]. They are also promising candidates for imaging [159], drug delivery [25,26], and antimicrobial activity [160]. In conclusion, while osmium nanoparticles are promising in addressing current therapeutic limitations in breast cancer treatment, more extensive research is essential to overcome existing challenges and pave the way for their applications in breast cancer therapy.

### 3.6. Iridium-Based Nanoparticles

Yuan et al. explored the potential of iridium dioxide nanoparticles (IrO_2_ NPs) combined with glucose oxidase (GOx) as an amplifier to enhance photodynamic therapy (PDT) and photothermal therapy (PTT) in breast cancer treatment (Figure 7a,b) [161]. PDT selectively destroys cancer cells [162,163,164]. However, PDT faces limitations such as insufficient photosensitisers and tumour hypoxia, which restrict its efficacy [161,165,166]. IrO_2_ NPs, with their photothermal/photodynamic effects and catalase-like activity, were synthesised and combined with GOx to form IrO_2_-GOx@HA NPs, which amplify PDT and PTT [161]. These nanoparticles target tumour sites by converting glucose into hydrogen peroxide via GOx, which is subsequently converted into oxygen by IrO_2_ NPs, thereby enhancing type II PDT. Additionally, IrO_2_ NPs exhibited photothermal properties for PTT [167]. Experimental results show a significantly higher photothermal conversion efficiency of 68.1% compared to MoO_2_ nanoparticles (62.1%). Under near-infrared light irradiation, IrO_2_-GOx@HA NPs reached a temperature of 66.5 °C within 10 min, outperforming MoO_2_-based counterparts in heat generation. In vitro and in vivo studies confirmed the efficacy of IrO_2_-GOx@HA NPs in enhancing PDT, inducing apoptosis in breast cancer cells (4T1), and alleviating tumour hypoxia [161]. The synergistic effect of PDT and PTT improved overall therapeutic outcomes.

IrO_2_ NPs demonstrated low cytotoxicity and high biocompatibility, making them suitable for biomedical applications [161]. Compared to other nanozymes, IrO_2_ NPs offer advantages such as high catalytic activity and stability [168]. In vitro experiments on cancer cell lines and in vivo studies using xenogeneic tumour models further validated the efficacy of IrO_2_-GOx@HA NPs in improving PDT therapy and combination cancer therapy [161]. The study suggested that IrO_2_-GOx@HA NPs hold promise for improving breast cancer therapy by overcoming the limitations of conventional treatments, with minimal toxicity and significant tumour inhibition observed in vivo.

Utilising photostable iridium(III)-cyanine complex nanoparticles (IrCy NPs) has shown promise in breast cancer therapy [169]. IrCy NPs, iridium nanoparticles with cyanine dye, have excellent near-infrared (NIR) absorption properties (Figure 8a). These nanoparticles exhibit high water solubility and photostability, making them suitable for various biomedical applications, including PDT [169]. In preclinical studies using breast cancer cell lines, IrCy NPs demonstrated effective photoacoustic imaging and the generation of singlet oxygen (^1^O_2_) upon 808 nm laser irradiation. This property enables real-time monitoring of pharmacokinetics, tumour targeting, and biodistribution of IrCy NPs in vivo. Photoacoustic imaging revealed a long blood circulation half-life of IrCy NPs, approximately 18 h, allowing for effective passive targeting of tumours via the EPR effect. The optimal time for in vivo therapy was 24 h post-injection, based on the highest photoacoustic signal intensity observed in tumour regions [169].

Biodistribution studies showed that IrCy NPs were partially excreted through the urinary system and underwent hepatic metabolism [169]. In vitro studies using 4T1 cells demonstrated the cytotoxic effects of IrCy NPs under laser irradiation, with significant reductions in cell viability observed. The ability of IrCy NPs to generate ^1^O_2_ was confirmed through fluorescence assays, highlighting their potential as effective photosensitisers for PDT. In vivo PDT experiments using a 4T1 xenograft mouse model demonstrated significant tumour ablation with minimal side effects. Tumour volumes were significantly reduced following PDT, and histological analysis confirmed treatment efficacy through apoptosis [169]. These findings suggest that IrCy NPs hold promise as a novel theranostic agent for breast cancer therapy.

Iridium (Ir) nanoparticles have shown potential in treating breast cancer due to their enhanced enzymatic activity and better tissue penetration in tumours due to their large surface area and small size. Wei et al. [107] investigated their role in breast cancer treatment; platinum group alloy nanoparticles of iridium and ruthenium (Ru) with GOx coupled on a PEG surface were synthesised. This study showed that IrRu-GOx@PEG nanoparticles inhibited the metastasis induced by hypoxia. IrRu-GOx@PEG nanoparticles also induced apoptosis in 4T1 cancer cells. The study revealed that the nanoparticle alloy treats breast cancer via oxidative and starvation therapy, where tumour growth is prevented by the termination of its nutrient source and the induction of apoptosis in cancer cells via catalysis of endogenous H_2_O_2_ to highly toxic ^1^O_2_, respectively [107]. The study further correlated with the findings on IrO_2_-GOx@HA nanoparticles, which enhanced type II PDT, inducing apoptosis in breast cancer (4T1) cells. The results also showed that Ir nanoparticles could address the limitations of current therapies, as IrO_2_-GOx@HA nanoparticles showed the potential to control and regulate tumour hypoxia, thereby improving the efficacy of PDT [161].

Furthermore, Zhang et al. [170] confirmed the findings of Ir’s role in PDT by synthesising diketopyrrolopyrrole (DPP) Ir nanoparticles and investigating their role in PDT in breast cancer cells. The study showed that incorporating Ir into DPP nanoparticles improved photothermal conversion efficiency from 42.1% to 67%. Additionally, DPP-Ir nanoparticles produced oxygen that could reverse the hypoxic tumour microenvironment, demonstrating their potential to regulate the tumour microenvironment [170].

Cyclometalated Ir(III) complex nanoparticles, such as Ir(tiq)_2_ppy, have demonstrated cytotoxic effects against MCF-7 cells through PDT. The mechanism of action of Ir(tiq)_2_ppy involves inducing caspase-activated apoptosis. This is significant as cyclometalated Ir(III) complexes are known for their use in PDT. However, by creating these complexes at the nanoscale, the challenge of poor water solubility is addressed, as the nanoparticles have shown enhanced water solubility and photoluminescence in aqueous solutions. Additionally, the cytotoxic effect of the nanoparticles was achieved at low concentrations and under weak light irradiation (1.6 μg/mL and 5 mW/cm^2^, respectively) [171]. This further demonstrates that nanotechnology can address the ongoing challenges faced by current therapeutic approaches. Yuan et al. [161] also investigated the role of iridium nanoparticles in PDT through a study involving IrO_2_ and GOx with hyaluronic acid (HA).

The ^192^Ir isotope is used in brachytherapy and plesio-brachytherapy for cervical and uterine cancer [29]. Iridium-coated nanoparticles are also used in dual-channel monitoring and detection, providing better sensitivity for the iridium signal in both in vitro tracking and in vivo models. Ir-NPs also disturb redox homeostasis in vitro, resulting in mitochondrial dysfunction and cell apoptosis. Ir-NPs exhibited low systemic dark toxicity, good tumour-targeting ability, and excellent antitumour effects. Additionally, Ir-NPs can be used as two-photon imaging agents for deep-tissue bioimaging (up to 300 μm depth). Multifunctional Ir-NPs show excellent prospects for bioimaging and PDT [172]. IrO_2_ NPs catalyse hydrogen peroxide (H_2_O_2_) in tumour microenvironments to generate endogenous oxygen, thereby enhancing PDT efficiency. Cellular and animal studies confirmed that the nanocomposites are biocompatible and have excellent tumour therapeutic effects. Iridium-based nanoenzymes hold potential for translational medicine [173].

IrO_2_ NPs can also act as thermosensitive PTT agents. In studies using 4T1 breast cancer cells, IrO_2_-GOx@HA NPs-mediated cell apoptosis and improved PTT outcomes [174]. In silico studies indicated that IrO_2_-GOx@HA NPs treatment could alleviate hypoxia inside tumour tissue, while in vivo studies further demonstrated that IrO_2_-GOx@HA NPs enhanced PDT efficacy. Ir NPs have shown promise in treating breast cancer effectively, and iridium-based complexes have exhibited anticancer activities with minimal side effects [174]. In vitro toxicity assays showed that cells tolerated IrNPs up to 10 μM iridium [175].

IrO_2_ showed excellent synergy in cancer cell treatment under near-infrared irradiation. In particular, in vivo therapeutic studies significantly inhibited tumour growth. Therefore, this research suggests a promising theranostic nanoprobe for tumour imaging and synergistic treatment of cancer cells [176]. Zhang et al. [177] demonstrated the anticancer potential of iridium compounds. Combined photodynamic and photothermal cancer therapy studies of iridium oxide showed good biocompatibility and achieved tumour-specific and enhanced combination therapy outcomes compared with the corresponding PTT or PDT monotherapy [178].

Iridium nanoparticles are recognised for their excellent biocompatibility and therapeutic potential. They are being explored for applications in imaging [179], aptasensors [180], biosensors [181], intracellular sensing [182], nanozymes, theranostic probes, and photosensitisers in PDT and PTT [182], as well as in drug delivery systems [182]. More possibilities for using iridium-based nanomaterials in breast cancer therapy can be explored. The preclinical studies of iridium and other PGM nanoparticles for breast cancer are summarised in Table 2. 

## 4. Conclusions and Future Perspectives

Research and development of PGMs in breast cancer therapy remain in the early stages (Table 2). However, PGMNs offer significant advantages, such as improved cancer targeting, enhanced imaging capabilities, better drug delivery, increased penetration, excellent physicochemical properties, high surface area, superior biocompatibility, and the potential for combinatorial therapies for diagnosis and treatment [183]. The reported successes in utilising PGMNs underscore the need to explore their applications in breast cancer therapy further. The deployment of PGMs could play a crucial role in reducing cancer mortality.

In conclusion, exploring PGMs in breast cancer therapy represents a promising new frontier in cancer research and treatment. Most PGMNs exhibit substantial antitumour activity against breast cancer cells while potentially minimising toxicity to normal cells. There is considerable potential for synergistic approaches to combat breast cancer, with minimal toxicity and significant tumour inhibition observed in preclinical studies. Therefore, further investigations into their mechanisms of action, optimisation of nanoparticle formulations, and translation into clinical applications are essential to realising their full therapeutic potential and improving outcomes for breast cancer patients.

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
