# Peer review of "Platinum Group Metals Nanoparticles in Breast Cancer Therapy"

_pharmaceutics, 2024, doi:10.3390/pharmaceutics16091162_

Round 1

Reviewer 1 Report

Comments and Suggestions for Authors

This manuscript basically meets the publishing requirements of Pharmaceutics in terms of content and format, but there are still some issues that need to be revised. I look forward to the author's revised version.

Major:

1.      Part 2 and part 3 can be combined.

2.      In introduction or conclusion part, lacking statement of unique advantages or characterizations of PGMN compared with other nanoparticles for cancer/breast cancer therapy. Especially in the conclusion, the author briefly introduced the wide range of biological applications of PGMN, which are all related to the main topic (I believe that’s unnecessary), but did not explain its anti-breast cancer properties.

3.      It is recommended that the author make a table that includes all the PGMN cases in Part 4. The columns in the table include the name of the nanoparticle, the metal contained, the cell lines treated, the therapy methods, the effect, the year, and others that the authors think is necessary.

4.      It is recommended that the author cite some high-level articles, such as ref 144 and the corresponding Fig. 8 (the authors introduced a lot), which are largely based on the further development of these original works: Angew. Chem. 2018, 130, 10466 –10470; Angew. Chem. Int. Ed. 2020, 59, 9491 – 9497.

Minor:

1.      The authors introduced the advantages of nanomedicine in cancer therapy in Fig. 1 and other paragraph, I’d recommend ref this paper: “Advantages of Nanomedicine in Cancer Therapy: A Review (ACS Appl. Nano Mater. 2023, 6, 24, 22594–22610)”, which systematically summarized all the advantage of nanomedicine for cancer therapy.

2.      Please give the full name before the abbreviations at the first time: HER2, FDA, EMA, PGMs; Many words that appear only once do not need to be listed in the abbreviation, such as PBS, FBS,

3.      Format of “drug delivery system” in Fig. 3.

4.      Make sure all “in vitro” or “in vivo” italic.

5.      Blank between number and unit.

6.      This paragraph is meaningless for the topic: PdNPs have excellent bacterial activity and are promising biosensors. Biosensors detect and target analytes, exploiting the high specificity of biological processes. PdNP-dopamine sensors boast high sensitivity and selection and a wide detection range [94].

7.      In part 4.2, the structure is mess, between examples, there’re some strange summary or nonrelative statements (eg, Pd nanoparticles have various promising biomedical applications [94]. Palladium nanoplates are useful in therapeutic gene loading and releasing in vitro combinational cancer therapy [97] Pd NPs served as drug delivery vehicles and delivered 131I, an anticancer radioisotope used in radiotherapy and doxorubicin (a chemotherapy drug). This produced synergists' anticancer effects from combining chem-, hotothermal-, and radiotherapy [98] using breast cancer cell lines (MCF-7)), please rearrange the structure.

8.      Figure 6 seems to be evidence against RuONPs and it’s blurry. I wonder why the author chose this figure?

Comments on the Quality of English Language

Long sentences have grammatical problems, such as the first sentence in the first paragraph of Part 2.

Reviewer 2 Report

Comments and Suggestions for Authors

1. In section 3, you need to provide a table that will systematize all clinical trials with nanoparticles in the treatment of breast cancer. The table should include the name of the drug, its active ingredient, clinical trial number, number of patients, details for which subtypes of breast cancer and at what stages the drug is being tested, result, patient enrollment/completed status.

2. For each of the metals under consideration, you also need to give a summary table of the compounds, on which cell lines it was tested, effects and their characteristics. This will make it easier to navigate the document.

Reviewer 3 Report

Comments and Suggestions for Authors

The manuscript entitled “Platinum Group Metals Nanoparticles in Breast Cancer Therapy” reviews in vitro and in vivo studies dealing with the biomedical use of nanoparticles based on metals of the platinum group focusing on the therapy of breast tumors.

Undoubtedly, the topic is interesting but some improvements in the text of the manuscript are required.

Specific comments

English should be improved as several sentences are unclear.

1.       Introduction – The meaning of the sentence “With developing countries experiencing a higher impact and current therapies having challenges from multi-drug resistance, enhanced toxicity, nephrotoxicity issues and low efficacy due to poor absorption rate, solubility, and delivery issues [9–11]” is unclear. Please, rephrase.

The meaning of the sentence “The use of nanoparticles addresses significant problems in current cancer therapies, such as the inability to detect cancer early and its toxicity” is unclear. What is the meaning of “to detect cancer early and its toxicity”?

2.       Section Nanotechnology in Breast Cancer Therapy – this section is confusing and should be better organized. The authors start the section talking about the potential of nanomedicine in the tumor treatment, then they insert the sentence “There are ongoing issues of having heavy metals in the body, and there are still concerns about using them in cancer therapy [21]” without explaining why metal nanoparticles could be useful in the treatment of tumors. Afterwards, the authors conclude “Therefore, the ideal PGMs must exhibit better therapeutic outcomes, improved bioavailability [22], prolonged survival, curbing side effects, reduced side effects and costs, and greener synthesis is preferred where possible”. After the above-mentioned sentences, the authors start talking about nanomaterials. Please, reorganize this section following a logical thread.

3.       Section Nanoparticles in Clinical Trials for Breast Cancer Therapy – The following sentences “There are very few nanomedicines available for clinical treatment of breast cancer. These nanomedicines include Nanoxel, Doxil®/Caelyx®, Lipusu®, Kadcyla®, Lipodox®, and Abraxane®” include information already reported in the previous section. When the authors describe the properties of Genexol®-PM, the type of nano-system used in this formulation should be specified.

4.       In the legend of Fig. 3, 5 and 8, the meaning of the abbreviations should be specified.

5.       Section Palladium-based Nanoparticles – The meaning of the abbreviation RGD should be specified. The meaning of the abbreviations should be specified the first time they are cited in the text. Please, check throughout the manuscript for all abbreviations. The meaning of the sentence “The highest cell percentage viability nanocomposite against normal (HUVEC) cell line was at 0 (μg/mL) concentration, while the highest was at 1000 (μg/mL) concentration” is unclear. Please, rephrase.

6.       Section Ruthenium-based Nanoparticles – The year “(2023)” reported after Mfengwana & Sone should be deleted. The sentence “Ruthenium and ruthenium oxide nanoparticles showed excellent prospects against breast cancer” at the end of the section is a repetition of a concept already reported in this section.

7.       Section   Rhodium-based Nanoparticles – the meaning of the abbreviation SPIO should be explained. The year “(2015)” after Chaves and co-workers should be deleted.

8.       Section Conclusion and Future Perspectives – This section is too long. The authors report a long list of citations that should be inserted in the appropriate sections. This section should be rewritten highlighting only the main outcomes and providing a critical analysis of the future perspectives.

Comments on the Quality of English Language

English should be carefully revised.

Round 2

Reviewer 1 Report

Comments and Suggestions for Authors

The author has made extensive revisions to the article and has basically solved my problem. I suggest that it be published after solving the following two minor problems:

1. Keep the format the same, eg., there are "Pt NPs", "PtNPs", "Pt-NPs" in Table 2.

2. Ref 28 seems not to be in the correct position, it should be around near ref 23.

Author Response

REVIEWER 

The author has made extensive revisions to the article and has basically solved my problem. I suggest that it be published after solving the following two minor problems.

Response: Thank you. We addressed the two minor revisions as mentioned below.

Comment 1:

  1. Keep the format the same, eg., there are "Pt NPs", "PtNPs", "Pt-NPs" in Table 2.

Response 1:

This was addressed (Pages 25 and 26; Table 2; 2nd Column). Platinum nanoparticles were abbreviated as Pt NPs.

Comment 2:

  1. Ref 28 seems not to be in the correct position, it should be around near ref 23.

Response 2:

Ref 28 was moved and is now reference 23. The reference list was updated accordingly.

Reviewer 2 Report

Comments and Suggestions for Authors

I have no more comments on the manuscript.

Author Response

Comment: I have no more comments on the manuscript.

Response: Thank you for the guidance and assistance.

Reviewer 3 Report

Comments and Suggestions for Authors

The authors revised the manuscript properly.

Comments on the Quality of English Language

Minor editing is required.

Author Response

Comment: The authors revised the manuscript properly. Minor editing is required.

Response: Thank you. We proofread and edited a few sentences to enhance the manuscript's cohesion and clarity.